



# OMPS-LP Aerosol Extinction Coefficients And Their Applicability in GloSSAC

Mahesh Kovilakam[1,2], Larry W. Thomason[2], Magali Verkerk[3], Thomas Aubry[3], and Travis N. Knepp[2]

[1]ADNET Systems Inc, MD, USA
[2]NASA Langley Research Center, Hampton, Virginia, USA
[3]Department of Earth and Environmental Sciences, University of Exeter, UK

**Correspondence:** Mahesh Kovilakam (mahesh.kovilakam@nasa.gov)

**Abstract.** The Global Space-based Stratospheric Aerosol Climatology (GloSSAC) is essential for understanding and modeling the climatic impacts of stratospheric aerosols. It primarily relies on data from the Stratospheric Aerosol Gas Experiment (SAGE) satellite series, supplemented by the Optical Spectrograph and Infrared Imaging System (OSIRIS) and the Cloud-Aerosol Lidar and Infrared Pathfinder Satellite Observations (CALIPSO). GloSSAC currently provides stratospheric aerosol extinction coefficients and aerosol optical depths at 525 and 1020 nm. With CALIPSO decommissioned and OSIRIS nearing the end of its operational life, SAGE III/ISS will soon become the sole data source for GloSSAC, but it will only be available as long as the International Space Station (ISS) is operational, around 2030. Therefore, incorporating other measurements, such as those from the Ozone Mapping and Profiler Suite limb profiler (OMPS-LP), is critical. OMPS-LP has provided continuous aerosol extinction coefficient measurements since 2012 with a retrieval algorithm developed by NASA (OMPS(NASA)). However, OMPS(NASA) has been shown to overestimate aerosol extinction coefficients, particularly after the 2022 Hunga Tonga eruption, compared to the tomographic retrieval of OMPS developed by the University of Saskatchewan (OMPS(SASK)) and SAGE III/ISS. Our analysis shows that OMPS(NASA) indeed exhibits a consistently high bias (>50%) following large volcanic eruptions and pyrocumulonimbus plumes from intense wildfires, while OMPS(SASK) shows reasonable agreement with SAGE III/ISS between $40^0$S and $40^0$N. This overestimation by OMPS(NASA) leads to an overestimation of the aerosol effective radiative forcing (ERF) and the associated model-simulated global surface temperature response by about a factor of two.

## 1 Introduction

Stratospheric aerosols play a crucial role in influencing the radiative and chemical equilibrium of Earth's atmosphere, in part due to their episodic enhancements associated with volcanic activity that can almost instantaneously enhance the total mass of aerosol in the stratosphere by several orders of magnitude (e.g., Thomason and Vernier, 2013). As a result, constraining stratospheric aerosol radiative forcing and understanding its climatic impacts is a major focus of many climate studies (e.g., Lacis et al., 1992; Myhre et al., 2013; Zanchettin et al., 2016; Timmreck et al., 2018; Krishnamohan et al., 2019). As a result, stratospheric aerosols are a key forcing in the 6th phase of the World Climate Research Programme's (WCRP) Coupled Model Intercomparison Project (CMIP6) (Eyring et al., 2016), as well as in the ongoing phase 7 (CMIP7). Most global climate models



(GCMs) participating in CMIP do not simulate the stratospheric aerosol life cycle interactively from emission; instead, they rely on prescribed aerosol optical properties based on global measurements (Stenchikov et al., 2006; Berdahl and Robock, 2013; Fyfe et al., 2013). GCMs with interactive stratospheric aerosols also often employ measurements to help evaluate their simulations (Aquila et al., 2013; Mills et al., 2016; Timmreck et al., 2018). To serve these needs, the Global Space-based Stratospheric Aerosol Climatology (GloSSAC) was first released in 2018 (Thomason et al., 2018). It is an outgrowth of the

SPARC Assessment of Stratospheric Properties (ASAP) and is an extensive multi-sensor climatology of stratospheric aerosol optical properties since 1979 that has been widely used by atmospheric modeling groups. It has played a crucial role in validating interactive aerosol schemes (e.g., Timmreck et al., 2018; Quaglia et al., 2023) and developing simple aerosol models that can be applied to provide aerosol optical properties to GCM without interactive schemes (Aubry et al., 2020). GloSSAC consists primarily of visible and near infrared aerosol extinction coefficient values in monthly, 5-degrees of latitude, and 0.5

km altitude grid that currently extends, as version 2.2, from 1979 through 2023 (NASA/LARC/SD/ASDC, 2024).

Stratospheric Aerosol and Gas Experiment (SAGE) measurements provide the backbone to GloSSAC and are available from 1979 to 1981 (SAGE), 1984 to 2005 (SAGE II) and 2017 to the present (SAGE III/ISS). Breaks in these time periods are filled with a combination of other space-based data, point-based data sets including lidar and balloon-borne instruments, and aircraft data (Thomason et al., 2018; Kovilakam et al., 2020). The 'filling' of missing data also includes the aftermath of

the Mt. Pinatubo eruption through where SAGE II observations were not possible in the lower stratosphere for several years. Improving this filling process remains the focus of ongoing research efforts. During the 2005-2017 gap, single wavelength measurements by OSIRIS and CALIOP were employed to fill the void (Thomason et al., 2018; Kovilakam et al., 2020). One disadvantage is that measurements of aerosol extinction coefficients at multiple wavelengths are crucial for inferring aerosol properties that is a primary application for GloSSAC data. Another issue is that the quality of even the single channel data

degrades during a number of small to moderate volcanic eruptions and intrusions of smoke during this period. While GloSSAC incorporates data from multiple sensors, its limitations must also be acknowledged. Solar occultation measurements, such as SAGE, are relatively straightforward measurements that are robust over a large range of stratosphere aerosol levels and provide near-global coverage. However, the measurement rate is slow and full zonal coverage requires about a month. In contrast, OSIRIS and CALIPSO offer daily near-global aerosol measurements opening a broad range of potential research

opportunities not provided by solar occultation. They are, however, limited in data quality by a retrieval process that relies on assumptions about aerosol microphysical properties, particularly size distribution and composition (e.g., Bourassa et al., 2012; Rieger et al., 2015, 2019; Kar et al., 2019). For the 2005-2017 gap in SAGE coverage and as a supplement in other periods, a conformance process has been implemented in GloSSAC (Kovilakam et al., 2020), serving as a mechanism to remove instrument-to-instrument bias. During periods when OSIRIS and CALIOP instruments have temporal overlap with a SAGE

instrument, this conformance process revealed that both data sets varied in their consistency when converting to the standard GloSSAC measurement wavelengths. This variation could be substantial and extend beyond variations associated with aerosol properties alone, particularly following volcanic and smoke events (Kovilakam et al., 2020).

With the end of the CALIPSO mission in 2023 and the impending end of OSIRIS's operational lifespan, only SAGE III/ISS will remain among the current ensemble of instruments used to create GloSSAC into the future. Realistically, even this in-



strument cannot be expected to survive much beyond 2030 or whenever the ISS is de-orbited ($\approx$ 2030). It is worth noting that there are currently no plans among the various space agencies to fly a solar occultation instrument in the future. As a result, our ability to extend GloSSAC into the future hinges on identifying a set of measurements that will be available that are of sufficient data quality and comparable spatial and temporal coverage. One obvious possibility is OMPS which has been operational since 2012 with additional flights sufficient to extend its observations into the foreseeable future. This data set

is available from two different groups using different retrieval approaches. OMPS(NASA) provides aerosol extinction coefficient retrievals at six wavelengths (510, 600, 675, 745, 869, and 997 nm) in version 2.1 (Taha et al., 2021). The University of Saskatchewan (SASK) offers a tomographic version of the OMPS retrieval, producing aerosol extinction coefficient data at only 745 nm. These data sets are generally in good agreement in background periods; however, large differences between the OMPS(NASA) and OMPS(SASK) products occur following events like the Hunga Tonga eruption in January 2022 (Bourassa

et al., 2023). Similar differences have been observed following other perturbations to the stratospheric aerosol levels including the 2019/2020 Australian bush fires. The Hunga Tonga event is of particular interest since it is one of the largest stratospheric events since that of Mt. Pinatubo in 1991 and because it is a period during which aerosol extinction coefficient measurements by OMPS, SAGE III/ISS, and OSIRIS exhibit differences.

Our goal in this paper is to examine and evaluate the two different products of OMPS data, OSIRIS and SAGE III/ISS

following the Hunga Tonga eruption and other stratospheric aerosol events since 2017. Ultimately, we aim to assess the best retrieval approach to integrate OMPS data into GloSSAC in the future without relying on constraining solar occultation observations. We will also assess the potential limitations of using these data sets to depict future volcanic and smoke events to understand the degree to which the GloSSAC data quality will be affected by this change.

## 2 Data and Methods

### 2.1 OMPS

The Ozone Mapping and Profiler Suite Limb Profiler (OMPS-LP) is a component of the Suomi National Polar-orbiting Partnership (Suomi NPP) satellite and the Joint Polar Satellite System (JPSS) satellites, launched in October 2011 (Flynn et al., 2006). While its primary focus is on monitoring the vertical structure of the ozone layer, OMPS-LP utilizes the limb-scatter technique to retrieve information about aerosols and related gases. Operating in a midday sun-synchronous orbit, OMPS provides

near-global coverage on a daily basis.

The retrieval methodology employed in OMPS-LP version 2.0, referred to as OMPS(NASA), involves establishing a one-to-one correspondence between a single limb radiance image and the retrieval of a singular aerosol extinction coefficient profile. This approach assumes horizontal homogeneity of the atmosphere along the line of sight. The retrieval is executed across six spectral bands at 510, 600, 675, 745, 869, and 997 nm (Taha et al., 2021). In the latest iteration (version 2.1), the incorporation

of more comprehensive convergence checks has improved the retrieval process, especially in the presence of volcanic plumes (Taha et al., 2022). Similar to other limb scatter measurements, the OMPS(NASA) procedure requires an assumption regarding the particle size distribution for deriving the scattering phase function. In the case of OMPS(NASA), a constant gamma function



size distribution derived from the Community Aerosol and Radiation Model for Atmospheres (CARMA) is utilized with $\alpha$ = 1.8, $\beta$=20.5, where $\alpha$ and $\beta$ are shape parameter and rate parameter respectively (Chen et al., 2018). We utilize version 2.1 of
OMPS(NASA) product for the analyses.

The OMPS LP instrument's imaging and rapid sampling capabilities facilitate the merging of consecutive limb images to create a two-dimensional retrieval of aerosol extinction coefficient. This method captures aerosol extinction coefficient simultaneously in altitude and along the orbit track. The University of Saskatchewan utilizes this ability to produce an alternative version of the OMPS data products, known as OMPS(SASK), in a tomographic retrieval approach. The limb scatter tomo-
graphic retrieval using OMPS-LP measurements was initially developed by Zawada et al. (2018). While this algorithm initially focused on ozone, it also incorporated the preliminary retrieval of aerosol extinction coefficient. The aerosol extinction coefficient is reported at 745 nm, featuring a vertical resolution of 1 km and a horizontal span of 125 km. Similar to other limb scatter retrievals, this algorithm operates based on assuming a particle size distribution, in this instance, employing a single-mode lognormal size distribution with $r_i$ = 0.08 and $\sigma$ = 1.6, where $r_i$ is the median radius and $\sigma$ is the distribution width. For
our study, we will utilize version 1.3 of OMPS(SASK). Another alternative version of OMPS data has been developed by IUP, University of Bremen (Rozanov et al., 2024) and may be considered for use in GloSSAC in the future.

## 2.2   OSIRIS

The Optical Spectrograph and InfraRed Imaging System (OSIRIS), a limb scatter instrument launched aboard the Odin satellite in 2001, retrieves $O_3$, $NO_2$, and aerosol extinction coefficient data (McLinden et al., 2012). Odin is positioned in a Sun-
synchronous orbit, providing coverage from $82^0$S to $82^0$N, excluding polar winter due to insufficient sunlight for measurements. The Optical Spectrograph instrument operates within the wavelength range of 284 to 810 nm with an approximate 1 nm resolution. A typical scan takes about 90 seconds, offering a vertical resolution of about 1 km, resulting in 100 to 400 profiles a day. The aerosol extinction coefficient retrieval is performed at 750 nm following a multiplicative relaxation technique, as detailed in Bourassa et al. (2012).
For GloSSAC, OSIRIS version 7.2 is currently used and plays a vital role in bridging the gap between the SAGE II period (1984-2005) and the SAGE III/ISS mission (June 2017-present). Significant improvements have been made to version 7.2 data (Rieger et al., 2019) compared to version 5.07 data (Bourassa et al., 2012). The retrieval in version 7.2 enhances the accuracy of the aerosol extinction coefficient product by minimizing the sensitivity to the unknown particle size distribution during inversion. This version aligns more closely with SAGE II and SAGE III/ISS compared to the previous version (v5.07)
in GloSSAC 1.0. While the mission is on-going, limitations to instrument operations have reduced the overall number of measurements made per orbit and concomitantly affect latitudinal coverage. Additionally, following Tonga eruption, OSIRIS aerosol extinction coefficient is about 50% less than those measured by SAGE III/ISS during this period. This discrepancy remains under study by the OSIRIS team[1].

---

[1]The OSIRIS team is actively investigating potential causes for underestimation, examining assumptions regarding size distribution in the retrieval algorithm and the impact of plume altitude on stray light (Adam Bourassa, personal communication).



### 2.3 SAGE III/ISS

SAGE III/ISS utilizes the solar occultation technique (McCormick et al., 1979), which measures the attenuation of solar
radiation caused by the scattering and absorption of atmospheric constituents during sunrise and sunset events. SAGE III/ISS
commenced data collection in June 2017 and represents an upgraded version of the SAGE III on Meteor (SAGE III/M3M)
instrument. Operating in a manner akin to its predecessors (e.g. Mauldin et al., 1985; Thomason et al., 2010), SAGE III/ISS
retrieves vertical profiles of multi-wavelength aerosol extinction coefficients (384, 449, 521, 602, 676, 756, 864, 1022, and

1544 nm), along with gas-phase species. The SAGE instrument family, renowned for its high precision (<5%), has a legacy
of providing vertical profiles of global stratospheric aerosols. These profiles serve as a benchmark for various correlative
measurements, facilitating comparisons and validation (e.g. Hervig and Deshler, 2002; Deshler et al., 2003, 2019; Rieger et al.,
2019; Bourassa et al., 2019). Due in part to a relatively straightforward process for inferring particularly mid-visible to near
infrared measurements of aerosol extinction coefficient, we consider observations by this instrument to be the standard against

which other approaches are evaluated. Furthermore, the SAGE series of measurements play a crucial role in creating GloSSAC
in conjunction with other space-based observations (Thomason et al., 2018; Kovilakam et al., 2020).

We have utilized version 5.3 of SAGE III/ISS data for all the analyses detailed in this paper. The modifications introduced in
version 5.3 are outlined in the release notes, available at https://asdc.larc.nasa.gov/documents/sageiii-iss/guide/ReleaseNotes_
G3B_v05.30.pdf. Notable changes in the solar product in version 5.3 encompass Disturbance Monitoring Package (DMP)

corrections specifically applied to solar products and a shift to using MERRA-2 72-layer data for meteorological input. While
SAGE III/ISS aerosol extinction coefficient measurements have been extensively employed for validation, comparison, and
long-term climatology purposes (e.g. Bourassa et al., 2019; Rieger et al., 2019; Kar et al., 2019; Kovilakam et al., 2020), a
negative bias has been observed in the aerosol channels (521, 602, and 676 nm) close to the Chappuis ozone absorption band
in the v5.2 aerosol data (Wang et al., 2020). Caution is advised when using these aerosol extinction coefficient measurements

at the above wavelengths due to the reported bias, and ongoing investigations are being conducted to address this issue.

### 2.4 Modeling of Effective Radiative Forcing and associated Surface Temperature Response

To assess the importance of differences among the aerosol optical property datasets considered in this study, we estimate the
global mean effective radiative forcing (ERF) and the associated temperature response for each dataset. To estimate the global
mean ERF, we use the relationship from Marshall et al. (2020):

$$\text{ERF} = -20.7 \times (1 - e^{-\Delta\text{SAOD}}) \tag{1}$$

where $\Delta$SAOD is the global mean 550 nm stratospheric aerosol optical depth (SAOD) anomaly. We calculate this anomaly as
the deviation of SAOD from its minimum over 2017-2023. Due to spatial coverage limitations and to preserve the temporal
resolution of the SAOD time series, we apply this relationship to the monthly $60^0$S-$60^0$N mean $\Delta$SAOD instead of the annual
global mean used by Marshall et al. (2020) for the calibration of Equation 1. We assume this approximation still provides a

reasonable estimate of the global mean ERF. Previous studies have typically assumed a linear relationship between ERF and





SAOD, with Forster et al. (2021) estimating a best value of -20 $\pm$ 5 W m$^{-2}$ per unit SAOD for the proportionality factor. Using a non-linear relationship better reflects the expected physical relationship between radiative forcing and SAOD and accurately captures the ERF-SAOD relationship in an extensive set of stratospheric aerosol injection simulations with the UM-UKCA interactive stratospheric aerosol model (Marshall et al., 2020).

To calculate the global mean surface temperature response associated with each stratospheric aerosol dataset, we use the Finite Amplitude Impulse Response (FaIR) model, a three-box impulse response model designed to mimic the behavior of complex Earth System Models (ESM) (Millar et al., 2017; Smith et al., 2018; Leach et al., 2021). This model transforms emissions of greenhouse gases and short-lived climate forcers into a concentration and radiative forcing time series, which is then used to estimate global temperature anomalies. We performed six 1000-member ensembles corresponding to the five

aerosol optical property datasets used in our study (OMPS(NASA), OMPS(SASK), OSIRIS, SAGE III/ISS, GloSSAC), plus an ensemble with no stratospheric aerosols-induced ERF. Each member is run with FaIR at monthly resolution with a different set of parameters (heat capacity, heat exchange coefficient for each box, climate feedback parameter, and forcing efficacy), effectively quantifying the climate modeling uncertainty. We run each ensemble with climate forcings from the Reduced Complexity Model Intercomparison Project (Nicholls et al., 2021), replacing the volcanic forcing time series with the ERF

time series calculated from Equation 1 using the SAOD time series corresponding to each aerosol optical property dataset tested. We do not propagate uncertainty related to climate variability, SAOD uncertainty, or SAOD-ERF conversion to keep the focus on differences between the stratospheric aerosol datasets and the expected mean temperature response.

## 3   Results

For consideration in GloSSAC, all datasets are required to be publicly available at a recognized data center and to have

undergone peer-reviewed validation for their stratospheric aerosol products, among other requirements (Thomason et al., 2018). Both OMPS(NASA) (Taha et al., 2021, 2022) and OMPS(SASK) (Bourassa et al., 2023) products satisfy these criteria, and are both valid, albeit competing, candidates for integration into the GloSSAC framework.

### 3.1   Evaluation of OMPS Aerosol Extinction Coefficient at 745 nm within the GloSSAC Framework

Given variations in measurement frequency and coverage across instruments, we utilize zonally averaged daily measure-

ments to facilitate meaningful comparisons. Our analysis focuses on aerosol extinction coefficient data at 745 nm from OMPS(NASA), OMPS(SASK), and OSIRIS at 750 nm, starting from January 2012 when OMPS data became available, extending to SAGE III/ISS measurements from June 2017 to the present. We opt to compare daily zonally averaged profiles from each instrument within a 4-degree latitude band, ensuring ample data for statistical analysis. Figure 1a illustrates daily zonally averaged aerosol extinction coefficient profiles for OMPS(NASA), OMPS(SASK), and OSIRIS, representing

a relatively unperturbed period (28 March 2013). The horizontal bars in the aerosol extinction coefficient profiles depict $\pm$ 1 standard deviation of the averaged profiles. The averaged aerosol extinction coefficient profiles for 28 March 2013 in the latitude band 30-34$^0$ N from each instrument exhibit good agreement above the tropopause, as shown in the percent difference





plot in Figure 1a. Percent differences remain mostly within $\pm$ 20% between the tropopause and 25 km, suggesting reasonable agreement between OMPS(NASA), OMPS(SASK), and OSIRIS for a period with relatively low aerosol levels. Notably,
differences within the scale of 20% are relevant to GloSSAC (Thomason et al., 2018; Kovilakam et al., 2020), provided the behavior of each instrument remains consistent and they vary systematically. Therefore, aerosol profiles during periods of low aerosol loading appear satisfactory within the context of GloSSAC. However, during perturbed periods, such as those caused by recent volcanic or smoke events, results may differ significantly, as observed following the Kelud volcanic eruption on 13 February 2014 at $8^0$S and the Calbuco volcanic eruption on 28 April 2015 at $41^0$S.

Figure 1b presents daily zonally averaged aerosol extinction coefficient profiles for OMPS(NASA), OMPS(SASK), and OSIRIS following the Kelud eruption. The averaged aerosol extinction coefficient profiles for 13 March 2014 in the latitude band $4^0$S-$2^0$N from each instrument reveal differences between the tropopause and approximately 23 km, where volcanic aerosol loading is evident. In this case, the difference not only exceeds the permissible difference criteria of $\pm$ 20% used for GloSSAC but also frequently exceeds 50%, and occassionally even more significantly, far surpassing acceptable bounds.
While the stratospheric aerosol optical depth (SAOD) from OMPS(NASA) and OMPS(SASK) does not exhibit significant differences, the SAOD difference between OMPS(NASA) and OSIRIS is notably distinct by more than 70%.



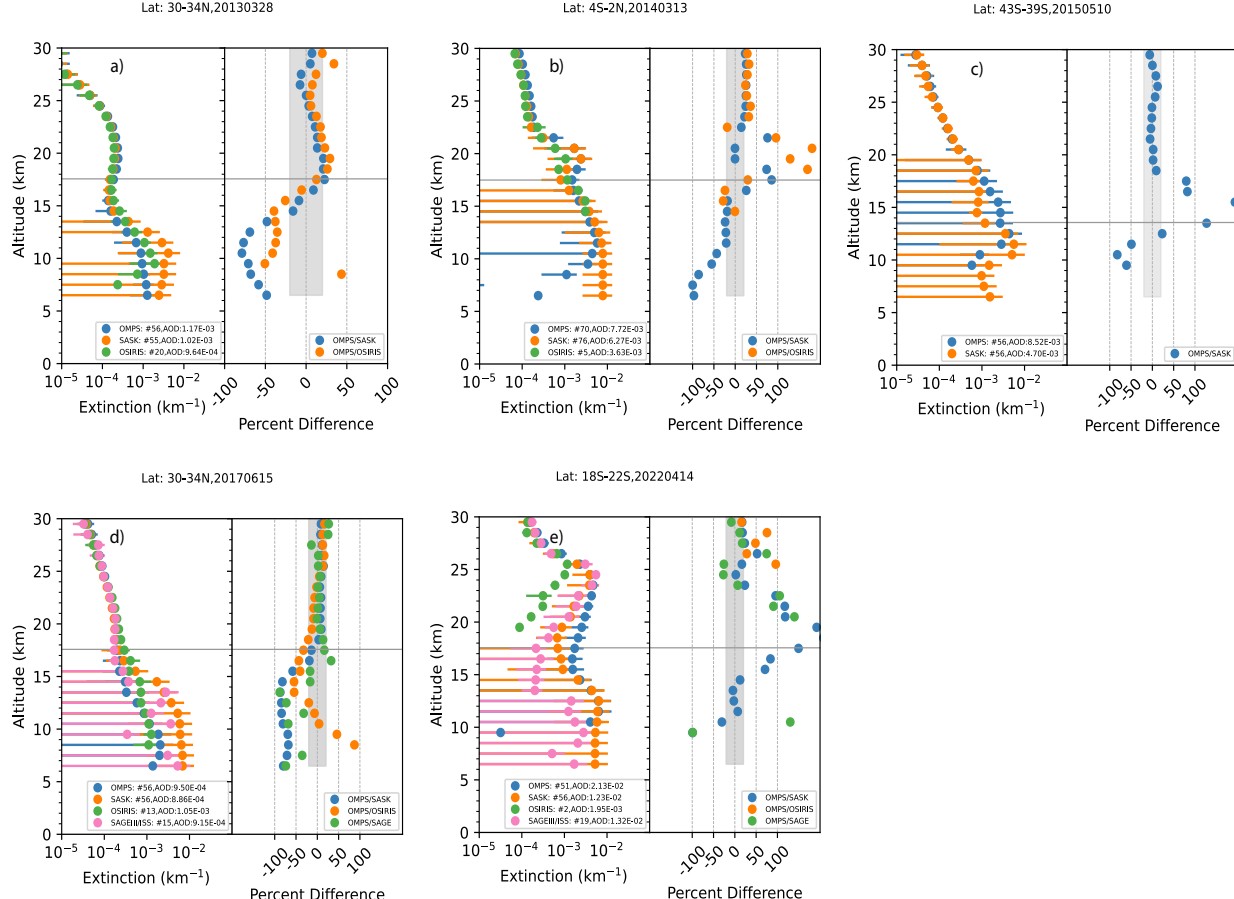

**Figure 1.** Daily zonally averaged profiles of aerosol extinction coefficient. OMPS(NASA) at 745 nm, OMPS(SASK) at 745 nm, along with OSIRIS at 750 and SAGE III/ISS at 756 nm, are used for comparison. The zonally averaged aerosol extinction coefficient is depicted for (a) the latitude band 30-34$^0$N for 28 March 2013 (relatively background stratosphere), (b) the latitude band 4$^0$S-2$^0$N for 13 April 2014 following the Kelud volcanic eruption on 13 February 2014, (c) the latitude band 43$^0$S-39$^0$S for 10 May 2015 following the Calbuco volcanic eruption on 28 April 2015, (d) the latitude band 30-34$^0$N for 15 June 2017 (relatively background stratosphere), and (e) the latitude band 18$^0$S-22$^0$S following Hunga Tonga eruption on January 15, 2022. The percent difference between OMPS (NASA) and other measurements for these events are shown in the adjacent plots as ((OMPS(NASA)-other)/other)*100, respectively. The gray shaded region in the percent difference plot indicates the permissible discrepancies between instruments ($\pm$ 20%) within the framework of GloSSAC. The error bars on aerosol extinction coefficient profiles represent aerosol extinction coefficient $\pm$ 1 standard deviation. The number of profiles used for averaging aerosol extinction coefficient profiles is also shown as #. The horizontal line represents the tropopause height. SAOD is computed by vertically integrating aerosol extinction coefficient between tropopause and 25 km.

Figure 1c illustrates daily zonally averaged aerosol extinction coefficient profiles following the Calbuco eruption for OMPS(NASA) and OMPS(SASK). The averaged aerosol extinction coefficient profiles for 10 April 2015 in the latitude band 43$^0$S-39$^0$S from





each instrument show differences between the tropopause and 18 km, where volcanic aerosol loading is evident. This contrast
is highlighted in Figure 1c, illustrating percent differences between OMPS and SASK, exceeding 70%. Furthermore, SAOD
from OMPS(NASA) significantly exceeds that of OMPS(SASK) by more than 50% as data between the tropopause and 18
km contributes more toward this difference as evident from the percent difference plots. No measurements are available from
OSIRIS for this comparison.

We also conducted an analysis of aerosol extinction coefficient data from OMPS(NASA) and OMPS(SASK) at 745 nm,
starting from June 2017, during which corresponding measurements were available from SAGE III/ISS. While a reasonable
agreement between OMPS(NASA), OMPS(SASK), and SAGE III/ISS (within $\pm$ 20%) is seen during relatively unperturbed
period in the stratosphere such as June 2017 (Figure 1d), the scenario is completely different following the volcanic eruption
of Hunga Tonga on January 15, 2022, at $21^0$S and the level of agreement among the instruments degrades dramatically. Fig-
ure 1e displays the zonally averaged aerosol extinction coefficient profiles for the latitude band $18^0$S-$22^0$S on April 14, 2022,
revealing that OMPS(NASA) significantly overestimates the aerosol extinction coefficient by approximately 80% or more rel-
ative to OMPS(SASK) and SAGE III/ISS. This finding corroborates the results of Bourassa et al. (2023). Notably, the percent
difference between OMPS(NASA) and SASK versions also exhibits significant differences, although not as those between
OMPS(NASA) and SAGE III/ISS. While overestimation in the lower stratosphere (approximately between the tropopause and
at about 20 km for the tropics) is a known issue for limb scatter measurements, mainly attributable to cloud contamination
at these altitudes (Rieger et al., 2015; Kovilakam et al., 2020), the percent difference of OMPS(NASA)/OMPS(SASK) and
OMPS(NASA)/SAGE III/ISS at altitudes between 25 and 20 km shows a clear overestimation within OMPS(NASA) data
set, exceeding 80% or more. Additionally, the computed SAOD between the tropopause and 25 km for the averaged aerosol
extinction coefficient profile (Figure 1e) shows that the OMPS(NASA) product (SAOD: 2.13 x $10^{-2}$) overestimated by ap-
proximately 60% compared to both OMPS(SASK) (SAOD: 1.23 x $10^{-2}$) and SAGE III/ISS (SAOD: 1.32 x $10^{-2}$), reinforcing
the fact that OMPS(NASA) overestimates aerosol extinction coefficient below the peak, in accordance with the findings of
Bourassa et al. (2023). Although, we present the result from OSIRIS here for comparison purposes, due to declining coverage
and low-frequency measurements in the southern hemisphere, OSIRIS only provides two profiles for this comparison and may
not properly represent aerosol profiles in the stratosphere following the eruption. Additionally, caution must be exercised while
using OSIRIS data after 2021 (Rozanov et al., 2024) due to underestimation of aerosol extinction coefficient.

In addition to the daily zonal averages, we also analyzed zonally averaged gridded data into 5 degree latitude bands. While
other eruptions such as Kelud, and Calbuco shows similar significant difference between OMPS(NASA) and other measure-
ments, we only show here an instance following Hunga Tonga eruption on January 15, 2022 for which all measurements
were available. Figure 2(a-d) illustrates zonally averaged aerosol extinction coefficient from OMPS(NASA), OMPS(SASK),
OSIRIS, and SAGE III/ISS for April 2022 following the Hunga Tonga eruption. Figure 2e distinctly shows a significant dif-
ference (OMPS(NASA) exceeding aerosol extinction coefficient by 50% or more) between OMPS(NASA) and OMPS(SASK)
in the latitude band $30^0$S to $30^0$N and for altitudes between the tropopause and about 24 km. Similar differences are observed
in Figure 2f between OMPS(NASA) and OSIRIS. Figure 2g also reveals a significant difference between OMPS(NASA) and
SAGE III/ISS. However, in addition to the overestimation of OMPS(NASA) aerosol extinction coefficient below the peak of




the aerosol layer at about 25 km, OMPS(NASA) underestimates aerosol extinction coefficient by about 50% at the peak of the

240 aerosol layer, around 25 km, within the latitude band $30^0$S and $30^0$N. Similar underestimation by OMPS(SASK) is observed when comparing OMPS(SASK) against SAGE III/ISS (not shown), suggesting that underestimation at the peak of the aerosol layer by OMPS retrieval, regardless of whether it is OMPS(NASA) or OMPS(SASK), is common for both OMPS products. This suggests that the underestimation could be attributable to the retrieval of OMPS aerosol extinction coefficient, where a fixed background size distribution assumption is used to derive the phase function (Taha et al., 2021; Bourassa et al., 2023).

245 The current analysis reveals a significant disparity between OMPS(NASA) and other datasets during essentially all perturbed events (eg., Kelud, Calbuco, and Tonga). Given the permissible difference of $\pm$ 20% between instruments in the context of GloSSAC framework (Thomason et al., 2018; Kovilakam et al., 2020), OMPS(NASA) significantly exceeds the bounds that are relevant to GloSSAC.

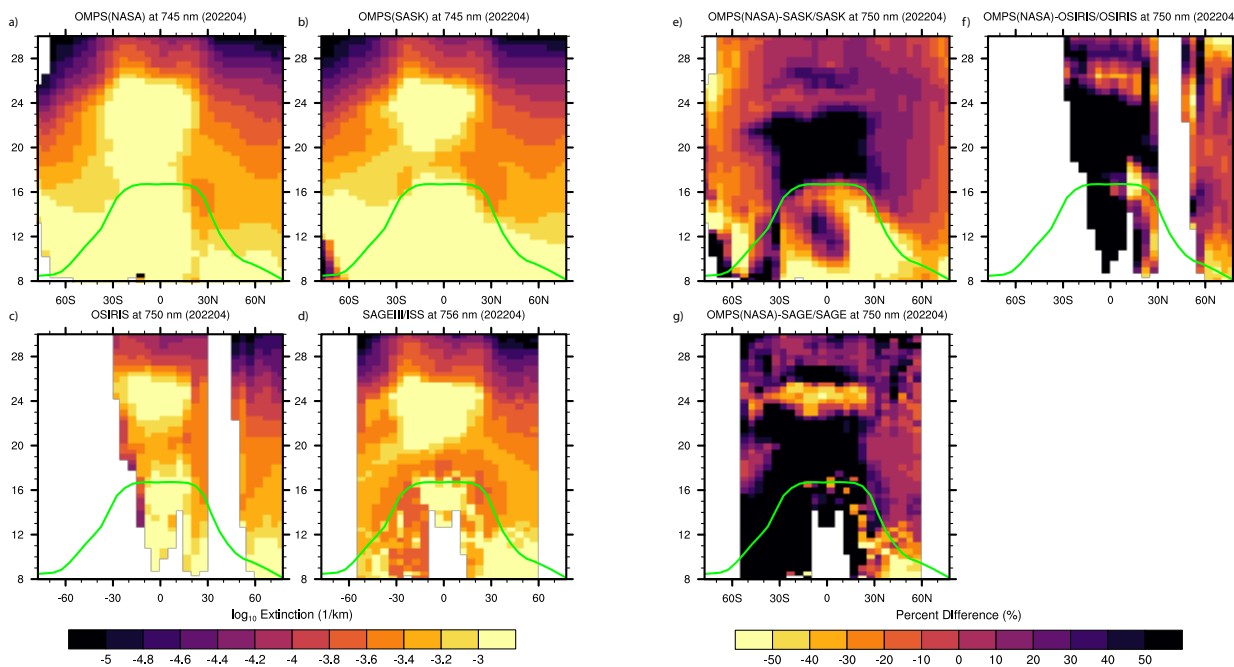

**Figure 2.** Zonally averaged gridded monthly aerosol extinction coefficient and percent difference for April 2022 following Hunga Tonga eruption on 15 April 2022 at $21^0$S. (a) Altitude versus latitude zonally averaged aerosol extinction coefficient from OMPS (NASA) at 745 nm, (b) same as in (a) but for OMPS(SASK), (c) same as in (a) but for OSIRIS at 750 nm, and (d) same as in (a) but for SAGE III/ISS at 756 nm. (d) shows the percent difference of zonally averaged aerosol extinction coefficient between OMPS(NASA)(a) and OMPS(SASK)(b), (e) illustrates the percent difference of zonally averaged aerosol extinction coefficient between OMPS(NASA)(a) and OSIRIS(c) and (f) shows percentage difference between OMPS(NASA) and SAGE III/ISS.The green solid line shows monthly averaged tropopoause from OMPS(SASK).

In addition to the aerosol extinction coefficient comparison, SAODs at 745 nm are also computed for the entire period

250 from 2012 to 2022. The SAOD was computed by integrating the aerosol extinction coefficient between the tropopause and



30 km using zonally averaged gridded data. Figure 3(a-d) presents time series plots of SAOD for OMPS(NASA) at 745 nm, OMPS(SASK) at 745 nm, OSIRIS at 750 nm, and SAGE III/ISS at 756 nm. All SAOD time series plots show an increase in SAOD following major volcanic/fire events. However, when examining the percent differences between SAOD values from different products, notable distinctions emerge. Figure 3e illustrates the percent difference between OMPS(NASA) and

OMPS(SASK), revealing significant overestimation of OMPS(NASA) aerosol optical depth following the Calbuco volcanic eruption in April 2015, Australian bushfire in January 2020, and the Hunga Tonga volcanic eruption in January 2022. These differences exceed 50% or more, persisting even 10 months after the Hunga Tonga eruption. While OMPS(SASK) demonstrates improved consistency following perturbed events, our analysis underscores the need for further evaluation, especially in instances of aerosol overestimation poleward of $40^0$S/$40^0$N in the SASK product. This discrepancy is likely attributed to cloud

contamination in the data, as evidenced by the overestimation of aerosol extinction coefficient in the lower stratosphere in the monthly zonally averaged gridded product (not shown). We anticipate that a more robust cloud filter could enhance accuracy in such cases. Additionally, a seasonal cycle is observed in the percent difference plot, particularly poleward of $40^0$S and $40^0$N, which could be attributed to changes in the scattering angle of OMPS. We also note significant overestimation of OMPS(SASK) between $20^0$S amd $20^0$N in the first couple of years of OMPS operations (Figure 3e), which could be attributed to changes in

the Level 1 OMPS product. The OMPS(SASK) team is currently investigating this issue, including the potential of moving the retrieval wavelength to 869 nm, which does not appear to have these issues (Adam Bourassa, personal communication).

A notable overestimation of OMPS(NASA) SAOD is also evident when computing percent differences in SAOD between OMPS(NASA) and OSIRIS. Figure 3f displays the time series of percent differences in stratospheric SAOD between OMPS(NASA) and OSIRIS. The plot clearly depicts an overestimation of OMPS(NASA) aerosol extinction coefficient fol-

lowing the Kelud eruption, Australian wildfires, and the Hunga Tonga eruption. OMPS(NASA) overestimates SAOD by approximately 50% or more following these events. OSIRIS aerosol extinction coefficient measurements during the Hunga Tonga time period exhibit a substantial low bias and deviate from patterns observed in previous events. The potential causes of this bias, including particle size assumptions, instrumental stray light, and sampling geometry, are currently under investigation (Adam Bourassa, personal communication). Therefore, while the overestimation of OMPS(NASA) aerosol extinction coeffi-

cient/SAOD following major events aligns with other products, the low bias from OSIRIS may have amplified the discrepancy between OMPS(NASA) and OSIRIS.

The SAOD from OMPS(NASA) at 745 nm tends to overestimate relative to SAGE III/ISS (Figure 3g), particularly poleward of $30^0$S, regardless of any volcanic event. Figure 3g also depicts a significant positive bias ($> 50\%$) in the southern tropics following Tonga eruption in January 2022. This discrepancy between OMPS(NASA) and SAGE III/ISS is in agreement

with previous studies (Bourassa et al., 2023). The results so far shows a clear discrepancy between OMPS(NASA) and other space-based measurements, reinforcing the fact that OMPS(NASA) aerosol extinction at 745 nm is beyond the permissible discrepancy limit ($\pm$ 20%) between the instruments, in the context of GloSSAC.







**Figure 3.** Startospheric SAOD time series. (a) Time series of stratospheric SAOD (Latitude versus time) for OMPS(NASA) at 745 nm, (b) same as in (a) but for OMPS(SASK), (c) same as in (a) but for OSIRIS at 750 nm, and (d) same as in (a) but for SAGE III/ISS at 756 nm. (d) shows the percent difference of SAOD time series between OMPS(NASA)(a) and OMPS(SASK)(b), (e) illustrates the percent difference of stratospheric SAOD between OMPS(NASA)(a) and OSIRIS(c) and (f) shows percentage difference between OMPS(NASA) and SAGE III/ISS. Major volcanic eruptions (white) and wild fire events (green) with abbreviated two-letter code with their respective latitude and time of occurrence that are listed here. The event names shown are Kelud (Ke), Calbuco (Cb), Canadian wildfire (Cw), Ambae (Am), Raikoke (Rk), Ulawun (Ul), Australian wildfire (Aw), California Creek Fire (Cc), La Soufriere (La), McKay Creek fire (Mc) and Hunga Tonga (Ht).





### 3.2 Multi-wavelength Aerosol Extinction Coefficients from OMPS(NASA) and SAGE III/ISS

Despite its lower frequency measurements compared to limb scatter instruments, SAGE III/ISS remains valuable with its
direct multi-wavelength aerosol extinction coefficient measurements, which provide valuable information about particle sizes,
particularly when the stratosphere is perturbed (Thomason et al., 2021). The OMPS(NASA) product also provides multi-
wavlength aerosol measurements, offering an opportunity to evaluate OMPS(NASA).

Figure 4 illustrates the time series of percent differences in SAOD between OMPS(NASA) and SAGE III/ISS at four wave-
lengths. While the 510 nm channel appears biased high throughout the entire time period (Figure 4a), regardless of any per-
turbed event, 745 nm shows better agreement (Figure 4b) relative to 510 nm. However, we note a consistent overestimation
($> 50\%$) of OMPS(NASA) at 745 nm poleward of $30^0$S. At 869 nm (Figure 4c), the differences are similar to 745 nm except
that the overestimation of OMPS(NASA) poleward of $30^0$S is improved relative to 745 nm. For 997 nm, Figure 4d depicts
a different pattern compared to Figure 4a, 4b, and 4c, showing that the overestimation only occurs following the Canadian
Wildfire and Hunga Tonga. Notably, 510, 745, 864, and 997 nm channels from OMPS(NASA) exhibit inconsistent behavior
across the spectrum, behaving differently in each channel relative to 521, 756, 864, and 1022 nm channels in SAGE III/ISS.





**Figure 4.** Startospheric SAOD time series. (a) shows the percent difference of SAOD time series between OMPS(NASA) and SAGE III at 510 nm, (b) the percent difference of SAOD time series between OMPS(NASA) and SAGE III at 745 nm, (c) the percent difference of SAOD time series between OMPS(NASA) and SAGE III at 864 nm and (c) the percent difference of SAOD time series between OMPS(NASA) and SAGE III at 997 nm. Note that for the percent difference we use 510, 745, 864, and 997 nm from OMPS(NASA) and 521, 756, 869, and 1022 nm from SAGE III/ISS. Major volcanic eruptions (white) and wild fire events (green) are same as in Figure 3.

In the context of GloSSAC, 510, and 997 nm from OMPS(NASA) and 521, and 1022 nm of SAGE III/ISS are particularly relevant. OMPS(NASA) product below 675 nm is not robust due to the low sensitivity of shorter wavelengths to aerosols (Taha et al., 2021). However, we use 510 nm for the comparison purposes with SAGE III/ISS as GloSSAC provides extinction coefficient at 525 nm, a channel close to OMPS(NASA)'s 510 nm and therefore relevant to evaluate OMPS(NASA) at 510 nm. For the analysis here, instead of examining extinction coefficient or SAOD, we examine stratospheric SAOD ratios between the channels from both OMPS(NASA) and SAGE III/ISS as extinction/SAOD ratios provide valuable information about the size of the aerosol particles (Thomason et al., 2021; Knepp et al., 2023; Kovilakam et al., 2023). For SAGE series



of measurements, 525 and 1020 nm extinction coefficient measurements are generally used for inferring particle size informa-
tion (Thomason et al., 2021; Knepp et al., 2023) and therefore, we evaluate OMPS(NASA) 510/997 nm SAOD ratios against

SAGE III/ISS 525/1020 nm SAOD ratio. Figure 5 depicts stratospheric SAOD ratio computed from SAGE III/ISS (Figure 5a)
and OMPS(NASA)(Figure 5b). An important difference between these two plots are the difference in the SAOD ratios. While
following each perturbed event, the SAOD ratio from SAGE (Figure 5a) clearly show changes in SAOD ratios, particularly
following Canadian wild fire, Raikoke eruption and Hunga Tonga eruption for which decrease in SAOD ratios are observed,
suggesting larger particles (Thomason et al., 2021; Knepp et al., 2023), whereas other moderate eruptions like Ambae and

Ulawun shows increase in SAOD ratios, suggesting increased amount of smaller particles. In contrast, OMPS(NASA) SAOD
ratios (Figure 5b) are completely inconsistent with SAGE III/ISS measured values and thus unlikely to be useful inferring any
correct information regarding aerosol size. The percent difference plot between SAGE and OMPS (Figure 5c) clearly suggests
that the OMPS(NASA) SAOD ratios are of significant bias ($> 50\%$), mostly due to the bias in 510 nm channel as mentioned
earlier and therefore are not suitable for GloSSAC purpose.





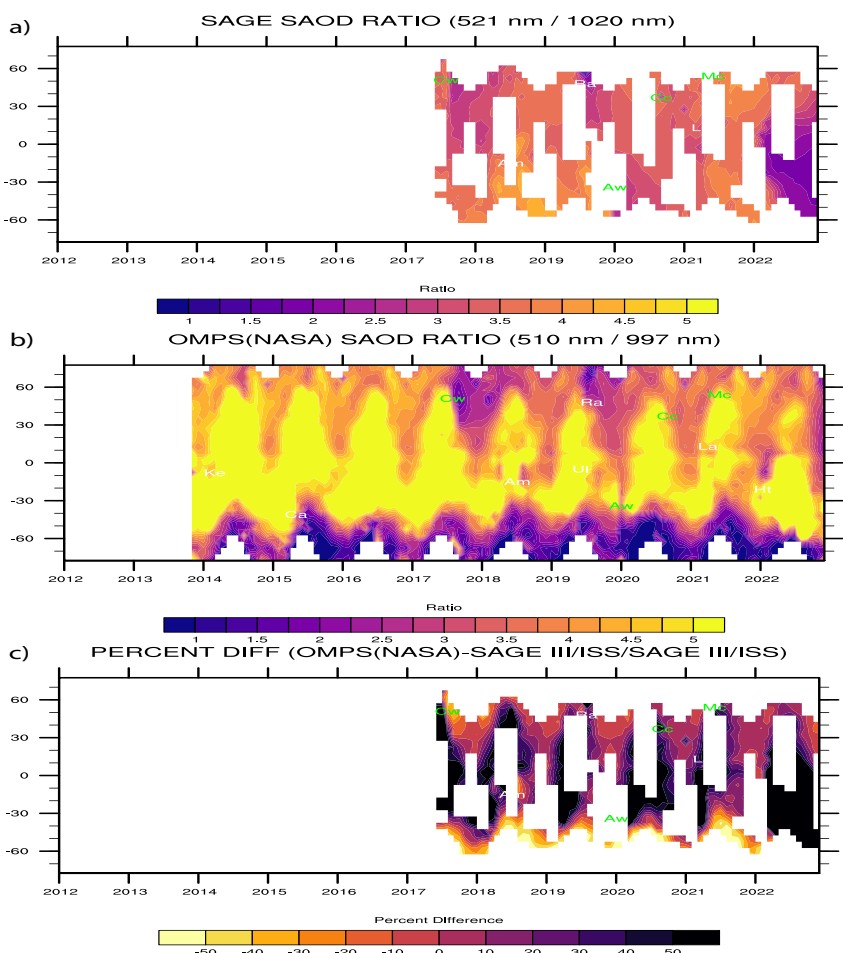

**Figure 5.** Startospheric SAOD ratio time series. (a) shows stratospheric SAOD ratio between 521 and 1020 nm from SAGE III/ISS, (b) shows stratospheric SAOD ratio between 510 and 997 nm from OMPS(NASA) , and (c) the percent difference of SAOD ratios between (a) SAGEIII/ISS and (b) OMPS(NASA). Major volcanic eruptions (white) and wild fire events (green) are same as in Figure 3.

From the aforementioned comparisons, it is evident that OMPS(NASA) tends to overestimate aerosol extinction coefficient and SAOD following major events in comparison to SAGE III/ISS and other space-based aerosol measurements. Moreover, the multi-wavelength measurements from OMPS at 510, 745, 869, and 997 are inconsistent across the channels, suggesting OMPS(NASA) multi-wavelength measurements lack to provide accurate information to infer particle size.

### 3.3   Evaluating OMPS(NASA) multi-wavelength measurements in the context of GloSSAC

For the purpose of comparison, we initially generated zonally averaged aerosol extinction coefficient profiles of OMPS(NASA) and gridded them to the same spatial resolution as GloSSAC (5-degree latitude resolution). Given that GloSSAC has a higher altitude resolution (0.5 km) than OMPS (1 km), we utilized GloSSAC at the OMPS altitude resolution to avoid any interpolation





along altitudes. While OSIRIS and CALIPSO have been components of GloSSAC version 2.22, the unavailability of CALIPSO data since January 2022 and the decreased coverage of OSIRIS resulted in GloSSAC relying predominantly on SAGE III/ISS

since January 2022. We utilized the publicly available GloSSAC version 2.22 data (NASA/LARC/SD/ASDC, 2024) for the analyses. In our analysis, we first selected data for June 2017, representing a relatively stable period, and another case following the Hunga Tonga eruption. The upper panel of Figure 6 (a, b) depicts the percentage difference of OMPS(NASA) and GloSSAC at 525 and 1020 nm for June 2017 for a relatively background stratosphere. While a reasonable agreement between OMPS(NASA) and GloSSAC was observed at 1020 nm (± 20%), except in the lower tropical stratosphere, significant discrep-

ancies between the two datasets persisted at 525 nm. Similar differences were noted between the tropopause and approximately 24 km in the tropics and southern mid-latitudes.

For the data following the Hunga Tonga eruption in January 2022, we conducted a similar comparison for April 2022 (Figure 6 c, d), by which time the volcanic plume had largely dispersed. We note that the overestimation of OMPS(NASA) persisted in both the 525 nm and 1020 nm channels between the tropopause and approximately 25 km. Interestingly, at the

peak of the aerosol layer, OMPS(NASA) underestimated GloSSAC by about 40%, while it overestimated below the peak of the aerosol layer by about 50% or more, exceeding the permissible limits (± 20%) between different data sets in the context of GloSSAC. The underestimation at the peak could be attributed to the fixed particle size distribution assumption, while the cause of the difference below the peak could be attributed to the difference in the retrieval algorithms between OMPS(NASA) and OMPS(SASK) and/or not converging the OMPS(NASA) product properly below the peak due to retrieval technique and

number of iterations.







**Figure 6.** Zonally monthly averaged percentage difference of OMPS(NASA) and GloSSAC (version 2.22) for 525 and 1020 nm on an altitude versus latitude plot. (a, b) for June 2017 and (c, d) for April 2022 following Hunga Tonga eruption.

In addition to evaluating aerosol extinction coefficient, we computed SAOD, as outlined in §3.1. Figure 7a, b displays latitude-versus-time plots of OMPS(NASA) SAOD at 510 and 997 nm, while Figure 7c, d illustrates that of GloSSAC 2.22 at 525 and 1020 nm. All plots reveal an enhancement in aerosol extinction coefficient following major events. However, the percent difference between OMPS(NASA) and GloSSAC at 525 nm (Figure 7e) shows overestimation of SAOD re-

gardless of volcanic events, and OMPS(NASA) consistently exhibits a high bias throughout the entire record. Therefore,





OMPS(NASA) aerosol extinction coefficient at 510 nm is not suitable for the GloSSAC purpose. The percent difference between OMPS(NASA) and GloSSAC at 1020 nm (Figure 7f) aligns better than Figure 7e. However, Figure 7f clearly shows overestimation of SAOD following Kelud, Calbuco, Canadian Wildfire, Australian Wildfire, and Hunga Tonga, suggesting that there is an inherent issue with OMPS(NASA), especially following perturbed events, and caution should be exercised in its

use.





**Figure 7.** Latitude versus time dependence of zonally averaged SAOD from OMPS(NASA) and GloSSAC (v 2.22). OMPS(NASA) SAODs are computed for 510 and 997 nm (a,b), while for GloSSAC SAODs are computed at 525 and 1020 nm (c, d). Percent difference between OMPS(NASA) and GloSSAC are also shown in (e) for 525 nm and (f) for 1020 nm. Major volcanic eruptions (white) and wild fire events (green) are same as in Figure 3.

Additionally, we computed the SAOD ratio between 510 and 997 nm of OMPS and compared it against SAOD ratio of GloSSAC between 525 and 1020 nm (Figure 8e, f). This comparison provides insights into how two wavelengths can provide information on aerosol extinction coefficient ratios, eventually aiding in the inference of aerosol particle sizes. Previous studies have demonstrated how aerosol extinction coefficient ratios can be leveraged to infer particle sizes using the SAGE series



of measurements (e.g., Thomason et al., 2021; Wrana et al., 2021; Knepp et al., 2023). We considered only the time series from 2017 through 2022 due to the availability of SAGE III/ISS multi-wavelength measurements from 2017. Figure 8e clearly shows that the SAOD ratios do not provide meaningful information, suggesting that these wavelengths cannot be used to infer any size information. However, on the other hand, the SAOD ratio from GloSSAC provides valuable insights (Figure 8f) into how ratios change following each volcanic/fire event, particularly after the Canadian Wildfire, Ambae, Ulawun, Raikoke,

Australian Wildfire, and Hunga Tonga. Each event behaves differently in terms of the SAOD ratio. For example, Canadian wildfire, Raikoke, Australian wildfire, and Hunga Tonga show a smaller SAOD ratio, suggesting relatively larger particles, while Ambae and Ulawun eruptions show a larger SAOD ratio, suggesting smaller particles. Therefore, it's crucial to emphasize that multi-wavelength aerosol extinction coefficients from solar occultation measurements, such as SAGE, is vital in providing important information on aerosol particle size-related details in GloSSAC.



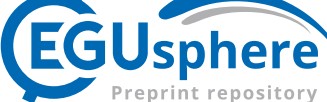

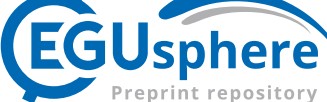

**Figure 8.** Latitude versus time dependence of zonally averaged stratospheric aerosol optical depth (SAOD) from OMPS(NASA) and GloS-SAC (v 2.22). OMPS(NASA) SAODs are computed for 510 and 997 nm (a,b), while for GloSSAC SAODs are computed at 525 and 1020 nm (c, d). Ratio between 510 and 997 nm of OMPS(NASA) SAOD is shown in (e), while (f) shows ratio between 525 and 1020 nm of GloSSAC 2.22 for the same time period. Major volcanic eruptions (white) and wild fire events (green) are same as in Figure 3.



### 4 Stratospheric Aerosol Effective Radiative Forcing and Associated Surface Temperature Response

Figure 9 shows $\Delta$SAOD averaged between $60^0$S-$60^0$N and corresponding global mean ERF estimates (see §2.4) for the period 2017-2022 $\Delta$SAOD from OMPS(NASA) is significantly different from other measurements, particularly following Tonga eruption in January 2022. The largest discrepancy occurs in August 2022, where the OMPS(NASA) SAOD peak is 0.036, whereas OMPS(SASK), SAGE III/ISS, GloSSAC, and OSIRIS SAOD values are approximately 0.013, 0.01, 0.01, and 0.006 respectively. On average, the $\Delta$SAOD difference between OMPS(NASA) and other measurements is about a factor of 3. Additionally, the discrepancy in OMPS(SASK) $\Delta$SAOD, relative to SAGE III/ISS and GloSSAC, is largely caused by an overestimation of extinction coefficient poleward of $40^0$S and $40^0$N as shown in Figure 3d. We then estimated ERF from the $\Delta$SAOD time series using Equation 1 (Marshall et al., 2020). In August 2022, at the peak of OMPS(NASA) $\Delta$SAOD, the OMPS(NASA) ERF of -0.73 W m$^{-2}$ is about a factor of 3.0 higher than SAGE III/ISS/GloSSAC as shown in Figure 9 b. This difference in ERF could be significant in terms of surface temperature impact (e.g., Ridley et al., 2014). Ridley et al. (2014) used a simple climate model to compute global volcanic aerosol forcing between 2000 and 2014, estimating it at -0.19 $\pm$ 0.09 W m$^{-2}$, which translates to an estimated global cooling of 0.05 to 0.12 $^0$ C.

We utilized the FaIR model to estimate the global mean surface temperature response from the estimated monthly ERF time series shown in Figure 9b (see §2.4). We obtain a peak cooling of 0.064 K induced by the OMPS(NASA) ERF, against a cooling of 0.021, 0.008, 0.018, and 0.017 K for OMPS(SASK), OSIRIS, SAGE III/ISS and GloSSAC respectively (Figure 9c). These results clearly indicate a cooling of about 0.092 K induced by OMPS(NASA) ERF, compared to cooling of 0.061, 0.039, 0.042, and 0.041 K for OMPS(SASK), OSIRIS, SAGE III/ISS and GloSSAC respectively, for August 2022. The global surface temperature continues to cool even months after the peak of OMPS(NASA) SAOD in August 2022, which is attributed to the higher heat capacity of the ocean resulting in a slow temperature response. For December 2022, we note a cooling of 0.105, 0.067, 0.042, 0.046, and 0.044 K for OMPS(NASA), OMPS(SASK), OSIRIS, SAGE III/ISS, and GloSSAC respectively. The temperature response from OMPS(NASA) exceeds a factor of two relative to SAGE III/ISS and GloSSAC.

For the Tonga eruption, we also calculated the difference between the peak cooling in December 2022 and the mean 2021 temperature response from Figure 9c. The OMPS(NASA) ERF induced a cooling of 0.064 K, compared a cooling of 0.021, 0.008, 0.018, and 0.017 K for OMPS(SASK), OSIRIS, SAGE III/ISS and GloSSAC respectively (Figure 9c). On average, over 2017-2022 time period, we estimate that SAOD forcing cooled climate by 0.031, 0.034, 0.022, 0.019 and 0.019 K on average in the OMPS(NASA), OMPS(SASK), OSIRIS, SAGE III/ISS and GloSSAC respectively. Thus, the OMPS(NASA) extinction coefficient product must be used with caution in stratospheric aerosol and climate studies, particularly following major volcanic events such as Tonga eruption.



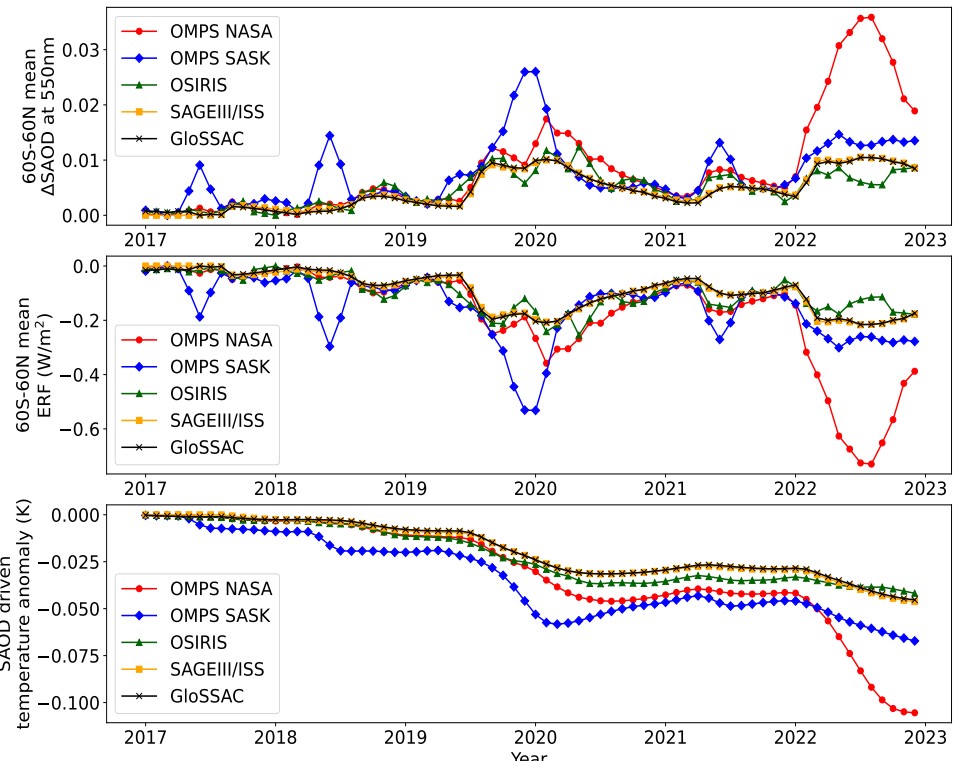

**Figure 9.** Time series of near global 550 nm $\Delta$SAOD (panel a), Effective Radiative Forcing (panel b), and temperature response (panel c) from OMPS(NASA), OMPS(SASK), OSIRIS, SAGEIII/ISS, and GloSSAC. To compute $\Delta$SAOD at 550 nm from 745 nm OMPS(NASA), OMPS(SASK), and OSIRIS, we used a constant Angstrom coefficient of 2.393, while for SAGE III/ISS and GloSSAC we used 1.97 to convert extinction from 525 to 550 nm. The resulting monthly surface temperature response in panel c is estimated from the FaIR model.

## 5    Summary and Conclusions

We have examined and assessed two publicly available OMPS aerosol extinction coefficient products namely OMPS(NASA) and OMPS(SASK) with other available space-based measurements within the context of GloSSAC framework. The analysis thus far revealed persistent disparities in the aerosol extinction coefficient, especially in the OMPS(NASA) product, demonstrating overestimation ($> 50\%$) following the Hunga Tonga eruption at 745 nm , which aligns with the findings of Bourassa et al. (2023). While Bourassa et al. (2023) focused on the period following Hunga Tonga and specifically at the 745 nm wave-

length, our analysis extends back to the year 2012, when OMPS began it's measurements. This extended time frame has enabled us to assess the consistent nature of these differences between OMPS(NASA) and other datasets, particularly in the context of volcanic eruptions/ Fire events. Furthermore, we leveraged this situation to evaluate OMPS(NASA) against OMPS(SASK), utilizing data from a couple of volcanic eruptions during that period, such as Kelud on February 13, 2014, at $8^0$S, and Calbuco



on April 22, 2015, at $41^0$S, and OMPS(NASA) is significantly overestimating ($> 50\%$) aerosol extinction coefficient for these
events as well.

In addition to the significant differences in OMPS(NASA) aerosol extinction coefficient at 745 nm, other wavelengths from
OMPS(NASA) also show inconsistency across different channels. As OMPS(NASA) provides aerosol extinction coefficient
data at multiple wavelengths, giving us the opportunity to compare the measurements at 510, 745, 869, and 997 nm with
those at 521, 756, 864, and 1020 nm from SAGE III/ISS. The comparison between OMPS 510 nm and SAGE III/ISS 521
nm reveals significant differences, with OMPS(NASA) consistently exhibiting a high bias ($> 50\%$) throughout the record,
regardless of any perturbed events. However, the overall agreement improves toward longer wavelengths. Despite this, overes-
timation of OMPS(NASA) aerosol extinction coefficient persists relative to SAGE III/ISS following major perturbed events,
showing a high bias in OMPS(NASA) measurements during periods of elevated stratospheric aerosol loading. Additionally, we
have computed SAOD from OMPS (NASA) and assessed with other products. The results clearly exhibit overestimation ($>$
$50\%$) of SAOD following major volcanic events. While the permissible differences between instruments used in the GloSSAC
framework are $\pm 20\%$ (Thomason et al., 2018; Kovilakam et al., 2020), the differences exceeding 50% or more in most of the
perturbed cases makes OMPS(NASA) product not suitable for GloSSAC.

Additionally, we assessed the consistency in OMPS(NASA) SAOD ratio between 510 and 997 nm compared to SAGE
III/ISS/GloSSAC SAOD ratio between 525 and 1020 nm for the period between 2017 and 2022, when SAGE III/ISS multi-
wavelength measurements are available. The OMPS SAOD ratios do not provide any meaningful information, suggesting that
510 and 997 nm wavelengths cannot be used to infer any size information (Figure 8e). In contrast, SAGEIII/ISS/GloSSAC
SAOD ratios exhibit consistency following each perturbed event, providing valuable insights into how ratios change following
each volcanic/fire event, particularly after the Canadian Wildfire, Ambae, Ulawun, Raikoke, Australian Wildfire, and Hunga
Tonga (Figure 8f). Moreover, aerosol extinction coefficients across different channels of OMPS(NASA) (510, 745, 869, and
997 nm) are not consistent, particularly following perturbed events. Therefore, it is crucial to emphasize that while limb scatter
measurements are valuable, they have not yet demonstrated the capacity to produce robust multi-wavelength aerosol extinction
measurements. They continue to produce lower quality data following volcanic and smoke events affecting the stratosphere and,
consequently, meaningful aerosol extinction coefficient ratios by which aerosol size information can be inferred. In contrast,
multi-wavelength aerosol extinction coefficient from solar occultation measurements, such as SAGE, plays a vital role in
providing important multi-wavelength information on aerosol particle size-related details in GloSSAC.

We also estimated SAOD-driven ERF from each instrument, finding that the OMPS(NASA) driven ERF has a larger impact
on radiative forcing compared to other datasets. At the peak of SAOD in August 2022, the OMPS(NASA) induced ERF is -0.73
W m$^{-2}$, which is about three times higher than the ERF from SAGEIII/ISS and GloSSAC (Figure 9b). We used the ERF time
series to estimate surface temperature response using the FaIR model. The results show a significant cooling of approximately
0.092 K induced by OMPS(NASA) ERF, compared to cooling of 0.061, 0.039, 0.042, and 0.041 K for OMPS(SASK), OSIRIS,
SAGE III/ISS and GloSSAC respectively, for August 2022 (Figure 9c).

Based on our analysis, the OMPS(NASA) product consistently exhibits a high bias (exceeding 50%), particularly in the
aftermath of perturbed events and in the GloSSAC context, the bias exceeds the allowable differences between the instru-



ments ($\pm$ 20%). In contrast, OMPS(SASK) demonstrates improved agreement with other space-based measurements. While
OMPS(SASK) aligns reasonably well (within $\pm$ 20%) with SAGE III/ISS at 745 nm, it overestimates extinction coefficient
poleward of $40^0$S and $40^0$N, potentially attributed to cloud contamination in the OMPS(SASK) data and a seasonal cycle.
Enhancing the cloud-clearing algorithm and potentially removing the seasonal cycle may address this issue and improve data
quality in these regions. Additionally, we note an overestimation of OMPS(SASK) extinction in the tropics in the first couple
of years of OMPS operations. Excluding the data quality issues in the first couple of years, the OMPS(SASK) product appears
usable in GloSSAC. However, challenges arise when converting OMPS-SASK aerosol extinction coefficient from 745 to 525
and 1020 nm, as GloSSAC provides aerosol extinction coefficient data at these wavelengths. Previous studies have employed a
constant Angstrom exponent for this conversion (Rieger et al., 2015), but this approach may not be suitable, especially consid-
ering the curvature in the aerosol spectrum following perturbed events such as volcanic eruptions or wildfires. To address this,
we plan to explore the potential of using a pseudo-Angstrom exponent, following the methodology developed in Kovilakam
et al. (2020, 2023). This process entails utilizing OMPS-SASK aerosol extinction coefficient at 745 nm in conjunction with
SAGE III/ISS aerosol extinction coefficient at 525 and 1020 nm to establish a monthly median pseudo-Angstrom exponent
climatology. Although employing a monthly median climatology proves effective, it may not sufficiently capture variations
in aerosol extinction coefficient at each wavelength, especially following a volcanic eruption or wildfire event, as highlighted
in Kovilakam et al. (2020). To address this limitation, we intend to refine the method by introducing a time-varying pseudo-
Angstrom exponent when both OMPS-SASK and SAGE III/ISS data are available. For data prior to June 2017, where we
did not have any SAGE measurements, we propose reverting to using a monthly median climatology of the pseudo-Angstrom
exponent.

In addition to the differences between OMPS(SASK) and SAGE III/ISS mentioned above, some differences between
OMPS(SASK) and SAGE III/ISS persist, especially at the peak of the aerosol layer following the Tonga eruption. Our fu-
ture work will include addressing this issue by investigating the sensitivity of the OMPS(SASK) algorithm to the particle size
distribution assumption, which is currently implemented as a constant lognormal size distribution. We plan to test the algorithm
with variable size distributions from SAGE III/ISS (e.g., Knepp et al., 2023; Ernest et al., 2024) to determine if this assump-
tion contributes to the underestimation of aerosol extinction coefficient at the peak of any enhanced aerosol layer. Preliminary
studies suggest that seeding the retrieval of limb-scatter measurements with size distributions inferred from solar occultation
measurements improves the quality of the limb-scatter instrument derived aerosol products. It may be possible to extend this
capacity to a post-occultation future but substantial and obvious hurdles will need to be overcome before this could be expected
to result in an acceptable data set for climate research and data sets like GloSSAC.

As previously highlighted, the SAGE series of measurements plays a pivotal role in GloSSAC, offering indispensable data
from 1979 to the present, albeit with a hiatus between August 2005 and May 2017. With the aim of developing stratospheric
aerosol properties for the upcoming phase 7 of the Coupled Model Intercomparison Project (CMIP7) forcing, GloSSAC has
been designated as the observational dataset, spanning the satellite era from 1979 onward. Therefore, the continued main-
tenance of the GloSSAC data record is essential, serving the crucial purpose of quantifying uncertainties in climate forcing
within CMIP7 and independently assessing the output of climate models.



*Data availability.* The GloSSAC v2.22 netCDF file is available from the NASA Atmospheric Data Center (https://asdc.larc.nasa.gov/data/ GloSSAC/GloSSAC_V2.22.nc) (NASA/LARC/SD/ASDC, 2024). The SAGE III/ISS and CALIOP data used in this study are available from NASA Atmospheric Data Center, while OSIRIS version 7.2 data are downloaded from https://research-groups.usask.ca/osiris.

*Author contributions.* MK and LWT developed the idea and methodology used in this paper. MK carried out all the analyses except for ERF and surface temperature analysis, which were performed by MV and TA, while TK participated in the scientific discussion. MK wrote the manuscript, while all authors reviewed the manuscript and provided advice on the manuscript and figures.

*Competing interests.* The authors declare that they have no conflict of interest.

*Acknowledgements.* We acknowledge the support of NASA Science Mission Directorate and the SAGE II and III/ISS mission teams. The SAGE mission is supported by the NASA Science Mission Directorate. ADNET personnel are supported through the RSES contract and NASA Grant 80NSSC 24K1185. MV acknowledges support from the faculty of environment, science and economy of the university of Exeter through a doctoral scholarship. We also acknowledge the International Space Science Institute (ISSI) team on " Perspectives on Stratospheric Aerosol Observations" for their valuable scientific discussions.



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
