# Peer review of "OMPS-LP Aerosol Extinction Coefficients And Their Applicability in GloSSAC"

_EGUsphere, 2024_

## Author Comment (AC1)

**Response to Reviewer #1**

We thank the reviewer for helpful comments. Our responses to the reviewer's specific comments are listed below. The reviewer's concerns are in bold italicized font and our responses are in regular font. The page numbers and line numbers given in our responses below are in reference to the revised version of the manuscript.

*The manuscript titled "OMPS-LP Aerosol Extinction Coefficients And Their Applicability in GloSSAC" by Kovilakam et al. evaluates and compares two different aerosol extinction coefficient products from the Ozone Mapping and Profiler Suite (OMPS) – OMPS(NASA) and OMPS(SASK) – along with other available space-based measurements (such as OSIRIS and SAGE III/ISS). The article aims to assess the performance of these products, particularly following significant stratospheric aerosol events like volcanic eruptions, and to determine the best retrieval approach for integrating OMPS data into the Global Space-based Stratospheric Aerosol Climatology (GloSSAC). Additionally, the article seeks to identify potential limitations of using these datasets for understanding future volcanic and smoke events and their impacts on climate. Their findings help inform decisions about which aerosol data products are suitable for integration into the GloSSAC, ensuring that the climatology remains robust and reliable. The article identifies potential limitations and areas for improvement in aerosol retrieval methods, guiding future research efforts to enhance data accuracy and reliability. It is recommended for publication after considering minor suggestions.*

**Specific comments**

*Page 1, L14, and here after: Update any instances of "400S" and "400N" to "$40^0S$" and "$40^0N$" throughout the document to accurately represent geographical coordinates using degree symbols.*

Done. Thanks.

*Page 9, L223, 224, Review and modify all occurrences of "2.13 X $10^{-2}$ and similar expressions. Replace them with "$2.13 \times 10^{-2}$" , ensuring consistent use of the multiplication symbol ($\times$) and superscript for exponents for clear scientific notation.*

Thanks. We have ensured consistent use of multiplication symbol ($\times$) and superscript for exponents in the manuscript.

*Page 23, L372, "overestimation of extinction coefficient poleward of $40^0S$ and $40^0N$ as shown in Figure 3d". Is it in Figure 3d? It seems the OMPS(SASK) data are shown in Figure 3b. If it is misreferenced, update it to refer to the correct figure.*

It is now corrected to Figure 3b (line 365). Thanks.

*And, it would be beneficial to expand on the discussion related to Figure 3, incorporating insights about the "overestimation of extinction coefficient poleward of $40^0$S and $40^0$N." This could involve explaining what the overestimation indicates and any relevant implications in this area.*

As noted in the manuscript (lines 265-269), the overestimation poleward of $40^0$S and $40^0$N may be due to cloud contamination in the OMPS(SASK) data and a seasonal cycle influenced by changes in scattering angle and assumptions about particle size. The OMPS(NASA) retrieval uses a gamma size distribution, while the OMPS(SASK) retrieval employs a log normal size distribution, which likely contributes to the differences in the seasonal cycles. We understand that the OMPS(SASK) team is currently investigating this issue and considering of moving the retrieval wavelength to 869 or 997 nm, which could help mitigate these problems.

*Page 24, check Figure 9 for panel markings. Add labels such as (a), (b), (c), etc., to each panel if they are not already present.*

Labels are now added. Thanks.

---

## Author Comment (AC2)

**Response to Reviewer #2**

We thank the reviewer for helpful comments. Our responses to the reviewer's specific comments are listed below. The reviewer's concerns are in bold italicized font and our responses are in regular font. The page numbers and line numbers given in our responses below are in reference to the revised version of the manuscript.

*** The manuscript presents a comprehensive comparison between aerosol extinction coefficients as derived from the OMPS-LP data set and other established data sets within the GloSSAC framework. In particular, the NASA product deriving the aerosol extinction coefficients at different wavelengths as well as a second product of the University of Saskatchewan (SASK) are utilized and compared to data from OSIRIS and SAGE III/ISS measurements. In conclusion, the OMPS-LP data exhibits a clear overestimation of the aerosol extinction coefficients compared to other data (exceeding ± 20%). It becomes clear from the manuscript that the NASA product cannot be used to benefit the GloSSAC framework, while there are ambitions to improve the SASK product and extend its retrieval to further wavelengths necessary to retrieve aerosol particle size information.
The discussions of the parameters presented are all comprehensive and well described. However, they slightly lack an overarching structure which could be overcome by inserting a brief outline at the end of the introduction summarizing the following studies.
The evaluation of the effective radiative forcing and associated surface temperature response helps to bring the previously discussed features into the context of climate modelling which is addressed again in the conclusions.***

**Specific comments**

*** In Section 3.2, the aerosol extinction coefficients at various wavelengths (510, 745, 864 and 997 nm) are assessed and an inconsistent behavior is found across the spectrum. Is this effect understood (particularly the deviation of the 997 nm data from the other data)? If so, please add an explanation to the text.***

We do not fully understand why the behavior improves at longer wavelengths. It is possible that the particle size (gamma distribution) sensitivity used in the NASA retrieval might align more effectively at longer wavelengths or that the increased aerosol to molecular ratio at longer wavelengths plays a role.

*** Section 3.3 evaluates the usability of OMPS(NASA) data in the context of the GloSSAC framework. The comparison in this section is limited to the time period of SAGE III measurements and thus partly redundant to the previous direct comparison of OMPS(NASA) and SAGE III data. I suggest to shorten this part considerably to***

*improve readability. This could be achieved by merging Fig. 7 and 8 and shortening the text to the necessary minimum; or moving the figures into an appendix and adding all relevant information in the text of the preceding section.*

Thanks. We have updated the time axis in Figure 7 to align with the SAGE III measurements period. Additionally, we merged the ratio plots into Figure7 as suggested. The text has been shortened to the essential points (lines 344-357).

**Technical Corrections**

*Please revise the text with special attention to the correct representation of units (e.g., usage of the degree sign $^0N$ and $^0S$), consistent naming (e.g., the Hunga Tonga eruption instead of Tonga eruption, OMPS(SASK) instead of SASK or OMPS-SASK) and the usage of articles (e.g., the aerosol extinction coefficient).*

We have now checked for consistency of units and the usage of articles. Thanks.

*l.47: stratosphere aerosol levels → stratospheric aerosol levels*

Corrected. Thanks.

*l.99: known as OMPS(SASK) → I suggest 'in the following referred to as OMPS(SASK)' since this is not an official abbreviation of the data set but a naming convention within this manuscript.*

Done. Thanks.

*l.230: 'zonally averaged gridded data into 5 degree latitude bands' is not completely clear to me. Please consider to extend the description of the shown parameter to enhance the understanding.*

We have now revised it to "In addition to the daily zonal averages, we also analyzed the zonally averaged aerosol extinction coefficient profiles from various instruments and gridded them to match GloSSAC's spatial resolution of 5 degrees latitude." (line 230-231)

*l.285: I suggest to write 'important information' instead of 'valuable information' to really stress the importance of direct multi-wavelength aerosol extinction coefficient measurements.*

Done. Thanks.

*l.374: As I understand the OMPS(NASA) ERF is - 0.73 $Wm^{-2}$ at its peak while the other data are at about -0.2 $Wm^{-2}$ which means that the number is by a factor 3 lower. Please consider rewriting 'a factor of 3 stronger' instead of 'higher'.*

Done. We have now changed it to "In August 2022, at the peak of OMPS(NASA) $\Delta$SAOD, the OMPS(NASA) ERF of -0.73 W m$^{-2}$ is about a factor of 3.0 stronger than SAGE III/ISS/GloSSAC" (line 366-367).

*l.380 and 389: You mention the numbers for the cooling twice. Please rephrase and avoid the repetition. Also, from the text it is not completely clear if you refer to the cooling at the peak ERF (August'22) or the peak cooling (December'22).*

We inadvertently put this sentence at line 372. We already stated this in line 380 as "The OMPS(NASA) ERF induced a cooling of 0.064 K, compared a cooling of 0.021, 0.008, 0.018, and 0.017 K for OMPS(SASK), OSIRIS, SAGE III/ISS and GloSSAC respectively (Figure 8c)." We therefore removed the sentence "We obtain a peak cooling of 0.064 K induced by the OMPS(NASA) ERF, against a cooling of 0.021, 0.008, 0.018, and 0.017 K for OMPS(SASK), OSIRIS, SAGE III/ISS and GloSSAC respectively (Figure 9c)." . Thanks.

*l.433: Same as in l. 374: the ERF is 'three times stronger' instead of 'higher'*

Done (line 425). Thanks.

*Fig. 1: Please enlarge the legends in each plot. Please consider adapting your color scheme. I suggest to use orange for the difference of OMPS to SASK and green for the difference of OMPS to OSIRIS and so on to match with the respective SASK and OSIRIS color in the extinction plots. You may also think about renaming OMPS $\rightarrow$ OMPS(NASA) and SASK $\rightarrow$ OMPS(SASK) to be consistent with the text.*

Done. Thanks.

*Fig. 2: Please ensure consistent labeling on the x-axis (60 $^0$S to 60 $^0$N versus -60 to +60) and add a label on the y-axis. For subplots e) to g) add the brackets (OMPS(NASA)-SASK)/SASK and again rethink consistent naming conventions.*

Done. Thanks.

*Fig. 3 to 8: Please correct the position of the brackets and add y-axis labels (see comment above). In Fig. 6 also the color bar label is missing as well as the exact description of the shown parameter in the figure caption.*

Done. Thanks.

*Fig. 8: You may think of adding the subplots e) and f) to Fig. 7 and skipping Fig. 8 because Fig. 8 a) to d) is just the same as Fig. 7 a) to d) with a modified time axis.*

Done. We have now merged the ratio plots with Figure 7 as suggested.

*Fig. 9: Please add the panel labels a), b) and c) in the plots or refer to it as the upper, middle and lower panel in the text.*

Done. Thanks.